# Survival Difference of Endometrial Cancer Patients with Peritoneal Metastasis Receiving Cytoreductive Surgery (CRS) with and without Hyperthermic Intraperitoneal Chemotherapy (HIPEC): A Systematic Review and Meta-Analysis

**DOI:** 10.3390/ijms25137495

**Published:** 2024-07-08

**Authors:** Ivan Panczel, Miklos Acs, Magdolna Herold, Viktor Madar-Dank, Pompiliu Piso, Hans Jürgen Schlitt, Magdolna Dank, Attila Marcell Szasz, Zoltan Herold

**Affiliations:** 1Division of Oncology, Department of Internal Medicine and Oncology, Semmelweis University, H-1083 Budapest, Hungary; ivan.panczel@gmail.com (I.P.);; 2Department of Surgery, University Medical Center Regensburg, D-93053 Regensburg, Germany; 3Department of Internal Medicine and Hematology, Semmelweis University, H-1088 Budapest, Hungary; 4Department of Finance, Rutgers University, Newark, NJ 07102, USA; 5Department of General and Visceral Surgery, Hospital Barmherzige Brüder, D-93049 Regensburg, Germany

**Keywords:** cytoreduction surgical procedures, endometrial neoplasms, hyperthermic intraperitoneal chemotherapy, peritoneal metastasis

## Abstract

Although several studies have been completed to investigate the effect of cytoreductive surgery (CRS) with or without hyperthermic intraperitoneal chemotherapy (HIPEC) in endometrial cancer with peritoneal metastasis (ECPM), a direct comparison was not performed previously. A meta-analysis was performed to investigate the suspected additional survival benefits of CRS plus HIPEC over CRS only. Twenty-one and ten studies with a total number of 1116 and 152 cases investigating CRS only and CRS plus HIPEC were identified, respectively. When all articles were analyzed, the 1-year survival rate was 17.60% higher for CRS plus HIPEC (82.28% vs. 64.68%; *p* = 0.0102). The same tendency was observed for the 2-year (56.07% vs. 36.95%; difference: 19.12%; *p* = 0.0014), but not for the 5-year (21.88% vs. 16.45%; difference: 5.43%; *p* = 0.3918) survival rates. The same clinical significance, but statistically less strong observations, could be made if only the studies published after 2010 were investigated (1-year survival rate: 12.08% and *p* = 0.0648; 2-year survival rate: 10.90% and *p* = 0.0988). CRS remains one of the core elements of ECPM treatment, but the addition of HIPEC to CRS can increase the positive clinical outcome, especially in the first 2 years.

## 1. Introduction

Endometrial cancer is the second most common gynecological cancer globally [1]. In the US, more than 60,000 newly diagnosed cases are registered annually [2]. A peak in the incidence of the disease is seen in those between the ages of 60 and 70, and there is a known strong correlation with obesity [3]. Histologically, more than 90 percent of uterine malignancies are of the endometrial type and originate from the epithelial layer. The remainder originate from mesenchymal cells and derive from the myometrial layer or the endometrial stroma [4]. The diagnosis of endometrial cancer in the early stage is usually favorable, with an overall 5-year survival rate of 85–95%, but the recurrence rate is very high (~10–15%) [5,6]. On the contrary, advanced stages (otherwise known as high-grade endometrial cancers) represent ~15% of newly diagnosed cases [7], which are associated with a significantly worse prognosis. In these cases, the cancerous cells are known to spread beyond locoregional points and metastasize to distant locations in the body, especially to the peritoneal cavity, including the gastrointestinal serosa. The 5-year survival rates are significantly reduced in women with metastases confined to the peritoneal cavity (from 20 to 25%), resulting in a median survival of less than 1 year [8].

Treatment options for endometrial cancer include surgical resection, systemic chemotherapy, brachytherapy, radiation, and hormone therapy. However, to date, the state-of-the-art treatment modality is a combination of surgical treatment and post-operative adjuvant chemotherapy [9]. In recent years, the treatment of endometrial cancer with peritoneal metastases (ECPM) has undergone a transformation, with the addition of cytoreductive surgery (CRS) with or without hyperthermic intraperitoneal chemotherapy (HIPEC). CRS aims to eradicate macroscopic tumors on the surfaces of the affected organs, whereas HIPEC introduces a heated chemotherapeutic agent into the peritoneal cavity in hopes of decreasing the microscopic malignant remnants [10]. The efficacy of CRS has been proven by multiple studies to be an effective treatment, not only for endometrial but also for multiple other disseminated cancer types [11]. Although CRS plus HIPEC in ECPM has been investigated by several studies, none of these compared the results to patients who had received CRS only. Therefore, a meta-analysis was performed, in which the aims were to shine a light on the advantages of the CRS-plus-HIPEC combination for the survival of ECPM patients, compared to CRS-only treated patients. In addition to the clinical comparisons, a further goal of this study was to conduct a systematic review of the molecular and/or cellular changes HIPEC induces.

## 2. Materials and Methods

### 2.1. Search Strategy

Following conceptualization, the meta-analysis was registered in the PROSPERO database (CRD42024510433). As the article presents aggregate data from previously published studies, ethical approval could be waived. The study was conducted following the Preferred Reporting Items for Systematic Reviews and Meta-Analyses (PRISMA) guidelines [12]. The search for eligible publications was performed in the BioMed Central (BMC), Cochrane Library, and the PubMed—Medical Literature Analysis and Retrieval System Online (MEDLINE) databases from their inception to 31 March 2024. The following search strings were used. The term “endometrial cancer” was combined with “cytoreductive surgery”, “CRS”, “hyperthermic intraperitoneal chemotherapy”, and “HIPEC” using the logical operator AND. Language restrictions were not used. No automation tool was used during the literature search. The literature search was conducted independently by two investigators (I.P. and Z.H.), and any discrepancies were resolved by consensus and, if necessary, by the opinion of a third and fourth reviewer (M.A. and A.M.S.).

Inclusion criteria for the studies were the use of CRS with or without HIPEC, FIGO stage IV [Fédération Internationale de Gynécologie et d’Obstétrique (International Federation of Gynecology and Obstetrics)] classification, and the inclusion of survival data, either in the form of survival rates, the number of deaths, or survival curves. Exclusion criteria included a short observation period (<1 year), if the patients treated with HIPEC could not be separated from the controls (mixed study groups), or if the study contained median survival and/or hazard rates only. Reviews, conference abstracts, in vitro and animal studies, and theoretical works were also excluded. 

### 2.2. Data Extraction

The following data were collected: the name of the author(s), year of publication, type of study (prospective or retrospective observational study, case series), sample size, mean age of patients, mean and standard deviation (SD) of the peritoneal carcinomatosis index (PCI) and hospital stay, elapsed time to HIPEC in months, and the 1-year, 2-year, and/or 5-year survival rates if available. If the authors of the published papers did not directly present the 1-/2-/5-year survival rate but the corresponding survival curve(s) of the cohort(s) was drawn, the percentage of patients alive at the specific time point(s) was read from the survival curve(s). Where individual data of the patients were presented only, the survival rates were manually calculated using the Kaplan–Meier method [13]. If the means and SDs of the collected parameters were not available in the original publications, the transformations published by Wan et al. [14], Luo et al. [15], and Shi et al. [16] were used [17].

### 2.3. Statistical Analysis

Statistical analyses were performed within the R for Windows version 4.3.3 environment (R Foundation for Statistical Computing, 2022, Vienna, Austria) using the R package meta (version 7.0-0) [18]. Survival rates were used for the effect size measurement, and random-effects models were performed. The heterogeneity variance measure (τ^2^) was estimated using the restricted maximum likelihood and Q profile method [19,20]. Between-study heterogeneity and publication bias were tested with Higgins and Thompson’s I^2^ statistic [21] and Egger’s regression test [22], respectively. Group comparisons were performed using the Mantel–Haenszel method [19,23,24]. Forest plots were used to graphically represent study results.

## 3. Results

### 3.1. Studies Investigating the Effect of CRS plus HIPEC on the Survival of ECPM

The electronic database searches for studies about the effect of CRS plus HIPEC on the survival of ECPM resulted in a total of 42 articles. No duplicates were identified, and after the removal of animal and cellular reports, reviews, and meeting/conference abstracts, 24 articles remained for title and abstract screening. In total, 7 of the 24 articles were excluded because they either reported results from other tumors or were review articles. Seventeen studies were considered for full-text assessment; however, a further seven of these had to be excluded due to the presence of different study interests (e.g., only hazard ratios from Cox regression models were included), the inclusion of other tumors, or their status as case reports (Figure 1).

Details of the ten CRS-plus-HIPEC studies selected for analysis can be read in Table 1 and Appendix A. Nine – nine [7,8,25,26,27,28,29,30,31] and five [7,26,28,31,32] of the selected studies contained information about 1-year, 2-year, and 5-year survival rates, respectively. For the 1-year survival rate analysis, the total number of patients included in the meta-analysis was 109, of whom 17 (15.6%) died during the first year after study inclusion. An 82.28% 1-year survival rate [95% confidence interval (CI): 70.99–93.57%] was estimated (Figure 2). It has to be mentioned, though, that some publications were present based on the results of Egger’s regression test (*p* = 0.0002). It was also investigated, using meta-regression methods, whether the effects of the publication years of the studies (*p* = 0.8296), the CRS-plus-HIPEC time (*p* = 0.7156), the elapsed time between disease diagnosis and HIPEC in months (*p* = 0.3421), the mean age of the patients (*p* = 0.9656), the mean peritoneal carcinomatosis index (PCI; *p* = 0.7377), or the mean hospital length in days (*p* = 0.7262) affected the analysis result. None of these parameters had any effect on the true effect sizes.

The 2-year and 5-year survival rates were also investigated. A total of 109 and 126 patients, and 48 (44.04%) and 94 (74.60%) deaths, were observed for the 2-year and 5-year data, respectively. A 56.07% (95% CI: 46.38–65.76%) 2-year survival rate and a 21.88% (95% CI: 10.40–33.35%) 5-year survival rate was estimated (Figure 3 and Figure 4). Neither the heterogeneity of the studies (2-year analysis: *p* = 0.3815; 5-year analysis: *p* = 0.6045) nor the investigated clinical parameters had an effect on the results of the analysis.

Most studies also included the perioperative PCI values and the length of the hospital stay. Eight of the studies contained detailed data about the perioperative PCI, and the average PCI throughout the studies was 12.40 (95% CI: 9.91–14.88; Appendix A). Nine of the ten studies detailed the hospital stay length, and its average was 16.22 days (95% CI: 10.56–21.88 days; Appendix A).

### 3.2. Studies Investigating the Effect of CRS Only on the Survival of ECPM

The literature search for CRS-only articles resulted in a total of 345 articles. After the removal of duplicates, animal and cellular studies, reviews, and meeting/conference abstracts, 209 articles remained for the title and abstract screening. A total of 158 articles were excluded due to their reporting about non-ECPM tumors, being a review or non-clinical publication, or because the full text was not available. The full texts were assessed for a total of 51 papers, from which an additional 29 had to be removed because they did not include sufficient statistical data to be included in the meta-analysis. Moreover, an additional paper [33] had to be excluded as it introduced a significant amount of publication bias due to its high number of cases (over 10,000), compared to the lower scale of the remaining studies. In total, 21 studies were included in the following analyses (Figure 5).

Details of the 21 CRS-only studies selected for analysis can be read in Table 2. It has to be noted that in contrast to what could be observed for the CRS-plus-HIPEC studies, somewhat fewer data were obtained for the CRS-only studies. Namely, the lengths of hospital stays and PCI data were barely detailed in any of the CRS-only studies. Of the CRS-only studies, 19, 20, and 18 studies with totals of 1006, 1068, and 982 cases were identified where details about the 1-year, 2-year, and 5-year survival rates could be found, respectively. Only one study [7] was identified that overlapped with the CRS-plus-HIPEC studies.

The 1-year survival rate of the ECPM patients was 64.68% (95% CI: 57.42–71.94%; Figure 6), and no publication bias was found (*p* = 0.5837). The 2-year survival rate was 36.95% (95% CI: 30.39–43.52%; publication bias: *p* = 0.4375; Figure 7), and the 5-year survival rate was 16.45% (95% CI: 11.74–21.17%; publication bias: *p* = 0.1025; Figure 8). It was also investigated whether the publication year or the age of the patients had any relationship with the true effect sizes. No association was found between the publication year and the effect size (1-year: *p* = 0.1405; 2-year: *p* = 0.2866, 5-year: *p* = 0.0751). The age of the ECPM patients had no effect on the 1-year survival rate (*p* = 0.6974); however, the 2-year [*p* = 0.0430, amount of heterogeneity accounted for (R^2^): 33.87%] and the 5-year (*p* = 0.0035; R^2^: 36.64%) survival rates were significantly affected by the older age of the patients.

Since both the surgical and oncological methods evolved significantly in the last 2–3 decades, it was also investigated whether the survival rates found in the studies conducted before or after the year 2010 differ. For the 1-year survival rates, a marginal difference was found [before 2010: 57.44% (95% CI: 44.00–70.88%); after 2010: 69.78% (95% CI: 66.07–73.48%); *p* = 0.0830; Appendix A]. The 2-year survival rates differed significantly [before 2010: 29.64% (95% CI: 20.72–38.56%); after 2010: 43.15% (95% CI: 35.27–51.03%); *p* = 0.0261; Appendix A], while no significant difference could be found in the 5-year survival rates [before 2010: 12.88% (95% CI: 7.16–18.59%); after 2010: 19.24% (95% CI: 12.12–26.36%); *p* = 0.1718; Appendix A].

### 3.3. Comparison of the Survival Differences between ECPM Patients Treated with CRS Only and CRS plus HIPEC

A direct comparison of the CRS-only and CRS-plus-HIPEC studies was also performed. First, all the available studies were compared, followed by the analysis of the papers published in 2010 or later. When investigating all available publications, the 1-year survival rate was significantly higher in the CRS-plus-HIPEC studies (CRS-only: 64.68%; CRS + HIPEC: 82.28%; difference: 17.60%; *p* = 0.0102; Appendix A; Table 3). Similarly, the 2-year survival rate was significantly higher in the CRS-plus-HIPEC studies (CRS-only: 36.95%; CRS + HIPEC: 56.07%; difference: 19.12%; *p* = 0.0014; Appendix A; Table 3). However, no statistical difference could be justified for the 5-year survival rates (CRS-only: 16.45%; CRS + HIPEC: 21.88%; difference: 5.43%; *p* = 0.3918; Appendix A; Table 3).

Thirteen and nine CRS-only and CRS-plus-HIPEC studies were organized after the year 2010, respectively. Similar results to those described in the previous paragraph were found. The 1-year (difference: 12.08%; *p* = 0.0648) and 2-year (difference: 10.90%; *p* = 0.0988) survival rates originating from the CRS-plus-HIPEC studies were clinically relevant but only marginally higher than the CRS-only results. The 5-year survival rates did not differ (difference: 2.64%; *p* = 0.7021; Table 3).

Although the following data could not be analyzed using meta-methods due to their heterogeneity/inconsistency, here we provide some additional information about the compared studies. It can be reported that the most common histology found was endometrioid adenocarcinoma (57%), followed by serous high-grade carcinoma (36%), carcinosarcoma (3%), and adenosquamous (0.4%) and clear cell carcinomas (3%). In terms of the adjuvant treatment, the most common post-operative therapy choice was chemotherapy with or without radiotherapy.

## 4. Discussion

Recurrence statistics of endometrial cancer are unfavorable, with early- (stages I and II) and advanced-stage disease (stages III and IV) showing 2–15% and 50% recurrence rates, respectively [54,55]. Peritoneal metastases are characteristic of both primary and recurrent endometrial carcinomas, and the life expectancy of patients presenting with such cases is extremely poor, usually with an estimated median survival of less than 12 months [8,56]. Such poor prognostic data has, in turn, led to major research and improvements in the treatment of endometrial cancers. Until recent years, the standard treatment for ECPM included CRS in combination with standard platinum-based chemotherapy only; however, the use of HIPEC in combination with CRS has lately spread as a more optimal method to treat disseminated endometrial cancers, amongst other malignancies [7]. CRS is an already well-established treatment modality with proven effects; e.g., Barlint et al. [57] have shown its benefits in a recent meta-analysis: overall survival (OS) ranged from 9 to 35 months in patients with advanced-stage or recurrent endometrial carcinomas. In contrast, HIPEC is a relatively new, innovative treatment modality, wherein specific chemotherapeutic agents are heated to 41–43 °C and are administered into the peritoneal cavity [10]. The aim of this treatment is to reduce the amount of microscopic, non-visible tumorous cells left on the peritoneal surface that cannot be removed surgically.

### 4.1. Cellular and Molecular Effects of HIPEC

The literature regarding the cellular and molecular mechanisms related to HIPEC is limited. Below is a summary of what is known on the topic. First and foremost, the process of heating a chemotherapeutic agent affects the activation of heat shock proteins and, in turn, not only induces the folding of intracellular proteins but also greatly modifies the resistance of tumor-involved cells to the pharmaceuticals. The induction of protein folding consequently results in the apoptosis of the affected cells [10,58]. For example, Kong et al. [59] have reported that when examining the effects of heated paclitaxel, the heat shock protein 27 (HSP27) levels were significantly downregulated, in contrast to the non-heated experimental condition. HSP27 is known to promote B-cell lymphoma 2 (Bcl-2) protein expression, inhibit Bcl-2-associated X (BAX) and caspase-3 expression, and reduce the BAX/Bcl-2 ratios. Thus, the use of HIPEC results in the inhibition of HSP27 expression, which indirectly promotes apoptosis in the affected tumor cells, ultimately leading to consequent anti-tumor outcomes [59].

Multiple studies have reported that increasing the temperature of the therapeutics enhances and intensifies the drug uptake of the cells located on the peritoneal surface, incurring faster and more optimal chemotherapeutic effects [10,58]. Alberts et al. [60] performed a clinical experiment using leukemia-model mice, wherein the hind legs of the animals were placed in a warm bath to imitate hyperthermia, and cisplatin was administered simultaneously. The inhibition of leukemia colonial formations was increased up to as much as twofold when the cisplatin was heated, compared to the control group, wherein the drug was administered at normal body temperatures [60]. Rietbroek et al. [61] have also proven the theory that heated chemotherapeutic agents show elevated levels of cytotoxicity. In their research, they have been able to show such effects in the case of oxaliplatin [61].

Peritoneal tumor deposits have limited vasculature and connection to the circulation; therefore, targeting these nodules by systemic chemotherapy is much more difficult [62]. One of the positive effects of HIPEC is that by administering the chemotherapeutic agent locally, a relatively high concentration can be reached in the peritoneal cavity while maintaining low plasma drug concentrations, and as a result, the risk of reaching toxic doses is reduced significantly, while the therapeutic effect of the drug increases significantly.

A recent study has examined the effects of HIPEC on human gastric-derived tumorous cells and found that the heating of 5-fluorouracil (5-FU) not only enhances the cytotoxicity of chemotherapeutics in gastric cells but also induces the cleavage of poly-adenosine diphosphate-ribose polymerase (PARP) and caspase-3, both of which take part in apoptosis. Moreover, the use of heated 5-FU decreased cell survival signals and chemoresistance-associated proteins and significantly elevated the expression of programmed death ligand 1 (PD-L1) on the surface of the tumor cells. Furthermore, it has also been reported that cells treated with heated 5-FU have higher levels of resistance toward recurrence [63]. The effects of HIPEC on the expression of microRNA in human gastric cells have also been examined, and it has been reported that microRNA-218 (miR-218) levels were upregulated significantly after the use of HIPEC. MiR-218 is known to elevate the chemosensitivity of gastric cancer cells to cisplatin in vitro. Moreover, the growth of the tumor cells has also been shown to be inhibited in vivo [64].

### 4.2. Clinical Effectiveness of CRS Only and CRS plus HIPEC in ECPM

#### 4.2.1. Studies Investigating CRS Only

Up until recent years, the treatment of not only peritoneally disseminated high-grade endometrial cancers but also other similar gynecological malignancies was solely based on the combination of cytoreductive surgery and (following that) platinum-based chemotherapy [65]. This entailed the surgical removal of all visible, macroscopic tumors from the peritoneal cavity, followed by systemic chemotherapy in hopes of providing the best possible prognostic outlook for such patients. The extent of the literature about the treatment of endometrial cancer by CRS without the use of HIPEC is large. In our meta-analysis, a total of 21 publications were eligible for inclusion and statistical analysis. After reviewing the articles, it can be summarized that, in general, all studies reported significant OS benefits in comparison to patients receiving only systemic chemotherapy. Our results showed that the estimated 1-year, 2-year, and 5-year survival rates of the ECPM patients treated with CRS were 65%, 37%, and 16%, respectively.

In the reviewed literature, the most significant effector that is associated with both improved OS and progression-free survival is the completeness of cytoreduction (CC) [34,49,66]. The CC scoring was proposed by Sugarbaker [67] to categorize how much of the visually detectable (macroscopic) tumor infiltration could be removed. Four categories were created: CC-0 is when no residual disease can be found after the completion of the procedure. CC-1, CC-2, and CC-3 are when the unresectable residual nodules measure less than 2.5 mm, between 2.5 mm and 2.5 cm, and greater than 2.5 cm, respectively [67]. In the current meta-analysis, the analysis of the comparison of optimal vs. suboptimal cytoreduction was not feasible. The reason behind this was that although the number of patients with various CC scores was indicated in almost all publications [7,34,35,36,37,39,40,41,42,43,44,46,47,48,49,51,52], only a few of the 21 CRS-only studies included this factor in any form of survival analyses [34,36,43,48,49,51]. Moreover, the definition of optimal and suboptimal cytoreduction significantly varied among the articles, which also prevented us from including these data for further analysis. The results of the articles can be generalized as follows. Optimal cytoreduction is associated with a significant survival benefit, compared to suboptimal cytoreduction. The median survival time of optimally cytoreduced patients is at least twice longer, sometimes even more, compared to that of the suboptimally cytoreduced patients.

The following parameters have also been associated with better survival in ECPM patients. Chemotherapy following surgery [35] and concomitant radiotherapy [37] have also been suggested as two of the significant effectors of longer survival (40 and 54 months, respectively). The histological subtype of the tumor [47], a recurrence in the pelvic site only, and a moderate or good ECOG performance status [44] are considered good prognostic indicators. However, the CA-125 level at presentation had no effect on OS [39]. It has to be highlighted that although multiple variables have been suggested by various authors as good prognostic factors, as listed earlier, no single, optimal treatment discipline has been determined yet. The latter can be attributed to several factors, but mainly to the significant heterogeneity of the studies. This heterogeneity comes, for example, from the fact that almost every study included both primary and recurrent cases. On the contrary, the treatment of ovarium tumors is now standardized, and after an international consensus was made, unified treatment regimens have been allocated [68]. There have been attempts to create a similar, gold-standard regimen to be used in the treatment of endometrial cancer. However, due to the lack of sufficient and/or appropriate evidence-based data, no such official guideline is yet available.

It was found during our analysis that the studies published after the year 2010 showed significantly better survival rates. This observation is most likely because CRS is becoming both more popular and significantly more researched as a treatment modality for a growing number of cancers. Moreover, new surgical and oncological developments have also been implemented over the years, which might have also influenced this observation. In the future, cytoreductive surgery is likely to undergo further modifications, both as a surgical method and as an oncological treatment, leading to even more optimal survival outcomes and, perhaps, a wider array of uses. With new therapeutic options on the horizon for ECPM, such as immunotherapy, it is likely that further modifications will arise in the treatment of this malignancy [69,70]. In conclusion, CRS as a treatment modality is both effective and safe when used to treat ECPM and coincides with improved survival statistics, as our meta-analysis also showed.

#### 4.2.2. Studies Investigating CRS plus HIPEC

CRS is an effective method for treating peritoneally disseminated high-grade endometrial carcinomas macroscopically; however, microscopic remnants cannot be treated in such a way. This has opened new possibilities in the treatment of ECPM, whereby chemotherapeutic agents are heated to 41–43 °C and inserted into the peritoneal cavity, a process better known as hyperthermic intraperitoneal chemotherapy (HIPEC). In comparison to that on CRS alone, significantly less literature exists on the use of CRS plus HIPEC in ECPM. In this meta-analysis, we could include 10 published studies with a total of 152 patients. It was found that the 1-year, 2-year, and 5-year survival rates of the patients receiving CRS plus HIPEC were 82%, 56%, and 22%, respectively. Similarly to what has been discussed in the case of CRS only, a major prognostic factor in the overall survival of patients treated with CRS plus HIPEC is the CC score [26,28,29,30]. Moreover, the recurrence-free survival (RFS) is longer if the CRS is completed with fewer surgical maneuvers, and if there are no lymph node metastases [32]. Furthermore, it was observed by Navarro-Barrios et al. [32] that those patients who did not receive neoadjuvant chemotherapy prior to CRS plus HIPEC had longer RFS times. This latter observation somewhat challenges our current knowledge that tumor burden is best reduced by neoadjuvant chemotherapy followed by CRS (+ HIPEC). It can be hypothesized that there may be cases wherein the chemotherapeutic agent is not effective enough, and the disease progresses despite treatment. Without precise knowledge of the individual cases, however, one can only speculate as to what might have been behind this observation. Therefore, further studies on this subject are required.

To date, no randomized clinical trial has been performed to assess HIPEC in ECPM, and only the results of observational studies are available in the literature. Therefore, a similar study to that of the PRODIGE-7 conducted for colorectal cancer [71] is needed in ECPM. It has to be noted, however, that the results of the PRODIGE-7 trial were not conclusive either, showing only a marginal improvement in the CRS-plus-HIPEC arm of the study [71]. In the current study, we identified several factors that ultimately led to an increase in the heterogeneity of the analyzed articles. Due to the fact that HIPEC is still evolving as a treatment modality, no consensus has been reached yet on subjects such as the optimal temperature to be used (41 °C vs. 42 °C vs. 43 °C), the length of HIPEC (60 vs. 90 min), and the chemotherapeutic agent used during the procedure. In all of the reviewed studies, the HIPEC temperature varied between 42 °C and 43 °C. The HIPEC length was either 60 or 90 min long, but there were studies in which the authors used both 60- and 90-min-long procedures. The chemotherapeutic agents used during the procedures also had a wide variability (cisplatin, doxorubicin, mitomycin C, and oxaliplatin). The latter increased the heterogeneity of the studies the most. These factors most likely contributed to most of the global discrepancies found between the different treatment regimens used, and to the inability to create a standardized, unified method. Landrum et al. have suggested that endometrial cancers should be treated similarly to ovarian cancers, both in terms of surgical and chemotherapeutic methods. However, molecular profiling has to be investigated further to optimize treatment [42].

#### 4.2.3. The Comparison of CRS-Only and CRS-plus-HIPEC Studies

To date, this is the first and only meta-analysis to summarize, review, and statistically analyze a large amount of literature regarding the effects of both the use of CRS alone and the combination of CRS with HIPEC on the overall survival of patients with peritoneal metastatic (stage IV) endometrial cancer. A significant 15–20% increase in the survival rates was found favoring CRS plus HIPEC in the first two years after the procedure, which decreased to below 10% for the fifth post-procedure year. It has to be mentioned that most of the HIPEC studies were conducted after the year 2010. Therefore, we analyzed our data with the removal of older CRS-only studies as well. In the latter analysis, we could observe a similar, clinically relevant but statistically less outstanding trend: in the first 2 years after the procedure, the HIPEC-treated patients had an increased survival rate of 10–15%, but there was no difference in the 5th year. The results observed in our study suggest the hypothesis that, with high probability, CRS plus HIPEC has a significant survival impact. However, this is difficult to prove due to the significant bias caused by the high heterogeneity of the observation-only studies available to date. It has to be mentioned that the observed survival difference is likely to increase with the recent introduction of immunotherapies, but there is no clinical data available yet, and we do not want to speculate. In other cancers, namely colorectal [72] and ovarian cancers [73,74], the use of HIPEC has also been reported to be advantageous. It has to be noted, though, that for recurrent ovarian cancer, the picture is not so clear as the PFS benefit is the same as the control groups’, warranting further research [73,74], similar to what we observed for ECPM.

To our knowledge, only a single study compared the two procedures in ECPM previously. In that study, Gomes David et al. [7] investigated 30 female patients who underwent CRS with or without HIPEC for the treatment of ECPM, respectively. Interestingly, in contrast to the results obtained in our meta-analysis, the median overall survival time was 19.2 months for the CRS-plus-HIPEC group and 29.7 months for the CRS-only group [7]. It has to be mentioned, though, that there were some major patient population differences in the study of Gomes David et al. [7], which could have led to the above-described inconsistencies. These include the fact that the CRS population was significantly closer to the date of diagnosis when the CRS was performed, while the CRS-plus-HIPEC procedure occurred significantly later. Until a randomized clinical trial rules out all doubts, such unfortunate inconsistencies in the literature mean that no coherence can be reached in the matter of the efficacy of using HIPEC as a supplementary treatment modality.

According to the current literature, the addition of HIPEC is not associated with significantly higher rates of complications [7,11]. Most of its side effects originate from the chemotherapeutic agent used. For example, the most common side effect of platinum-containing intraperitoneal chemotherapy is renal toxicity, which can be avoided by the use of sodium thiosulfate. Furthermore, hematologic side effects, such as the inhibition of hematopoiesis and neutropenia due to bone marrow suppression, are common after HIPEC [75]. Nevertheless, most of the complications are more likely to be attributed to the extensive cytoreduction and not to the additive HIPEC. In a recent multi-institutional comparative study from the PSOGI and RENAPE groups [7], where 30 patients were compared (CRS with HIPEC vs. CRS alone), the Grade III and IV complications according to the Clavien–Dindo classification rates did not significantly differ between the CRS-plus-HIPEC and CRS-only groups (20.7% vs. 20.7%, *p* = 0.739). In this study, one patient (3.3%) in each group experienced an abdominal hemorrhage and required a blood transfusion. The most frequent complication was gastrointestinal complication (overall, 11.8% of patients), which was not further specified. Another study conducted by Cornali et al. [28] reported Clavien–Dindo I-II, III, IV, and V complication rates for 33.3%, 15.1%, 3%, and 3% of the 33 included patients, respectively.

### 4.3. Strengths and Limitations of the Meta-Analysis

To the best of our knowledge, the current study is the first to analyze CRS-only and CRS-plus-HIPEC study results of ECPM patients in a single meta-analysis. However, a few limitations of this analysis should be mentioned. For instance, only a limited number of trials could be investigated, and all of them were non-randomized trials. The heterogeneity of the included studies was high, which might have also introduced some bias. Most heterogeneity arose from the use of various chemotherapeutic agents, and from the fact that both primary and recurrent cases were included in the studies. Moreover, the presence of lymph node metastasis and its effect on patient survival was often omitted by the original studies. Another limitation of the current study was that the numbers of available studies investigating CRS only and CRS plus HIPEC were significantly different, which is because, up to now, the number of surgical centers performing HIPEC has been very limited around the world, as opposed to those performing CRS. This, ultimately, might have affected the calculated pooled effect sizes.

## 5. Conclusions

The treatment of advanced endometrial cancer is currently evolving, with the good response to immunotherapy opening up a new horizon [69,70]; its use in combination with HIPEC is also feasible [76,77]. Nevertheless, surgical therapy also remains a core element of treatment in both primary and selected recurrent patients. In the context of surgical treatment using CRS plus HIPEC compared to using CRS alone, there appears to be a clinically important survival advantage when using the additive intra-abdominal hyperthermic chemotherapy based on the current meta-analysis. However, the results are somewhat overshadowed by considerable heterogeneity of the examined studies. Regarding the most important possible variables, an international consensus identical to that for ovarian cancer [68] should be reached in the future.

## Figures and Tables

**Figure 1 ijms-25-07495-f001:**
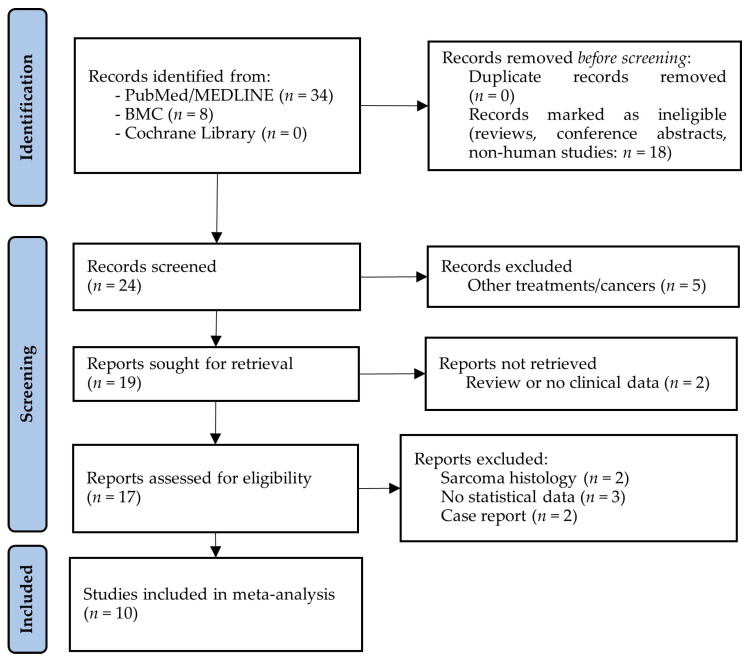
PRISMA flow diagram of studies investigating the effect of cytoreductive surgery (CRS) with hyperthermic intraperitoneal chemotherapy (HIPEC) on the survival of endometrial cancer patients with peritoneal metastases. BMC: BioMed Central; MEDLINE: Medical Literature Analysis and Retrieval System Online.

**Figure 2 ijms-25-07495-f002:**
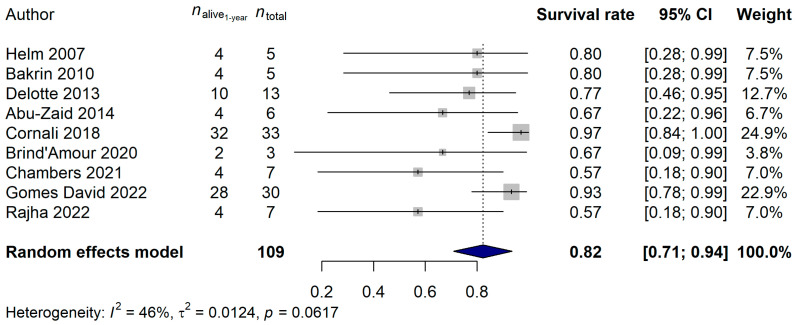
Effect of cytoreductive surgery (CRS) with hyperthermic intraperitoneal chemotherapy (HIPEC) on the 1-year survival rate of endometrial cancer patients with peritoneal metastasis. Cited references: [7,8,25,26,27,28,29,30,31]. Whiskers, dotted line and blue diamond represents the 1-year survival rate and its 95% confidence interval (CI), the computed effect size, and the latter’s CI, respectively.

**Figure 3 ijms-25-07495-f003:**
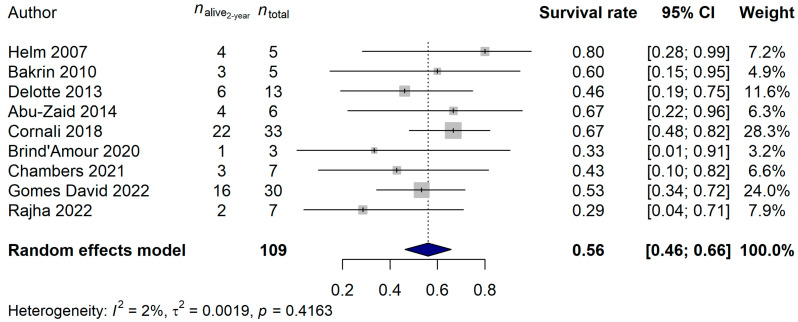
Effect of cytoreductive surgery (CRS) with hyperthermic intraperitoneal chemotherapy (HIPEC) on the 2-year survival rate of endometrial cancer patients with peritoneal metastasis. Cited references: [7,8,25,26,27,28,29,30,31]. Whiskers, dotted line and blue diamond represents the 2-year survival rate and its 95% confidence interval (CI), the computed effect size, and the latter’s CI, respectively.

**Figure 4 ijms-25-07495-f004:**
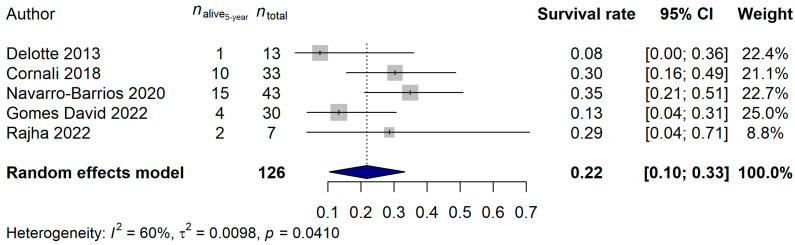
Effect of cytoreductive surgery (CRS) with hyperthermic intraperitoneal chemotherapy (HIPEC) on the 5-year survival rate of endometrial cancer patients with peritoneal metastasis. Cited references: [7,26,28,31,32]. Whiskers, dotted line and blue diamond represents the 5-year survival rate and its 95% confidence interval (CI), the computed effect size, and the latter’s CI, respectively.

**Figure 5 ijms-25-07495-f005:**
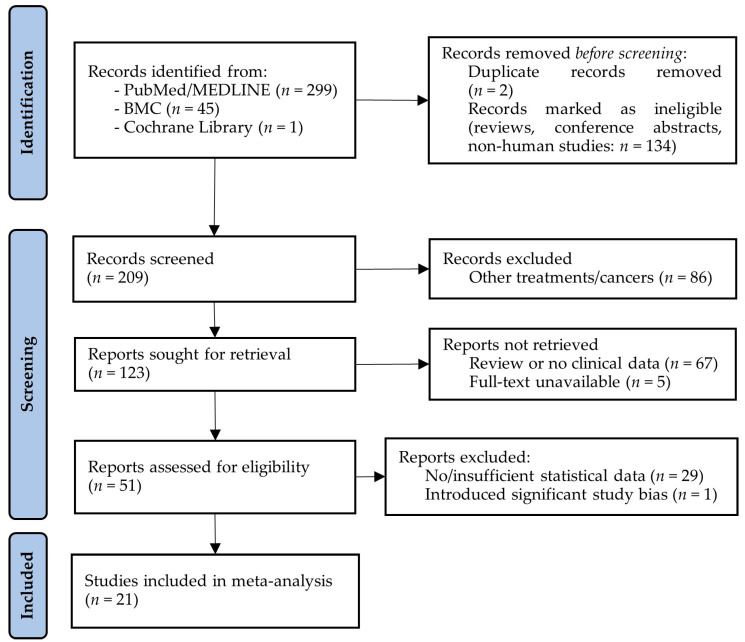
PRISMA flow diagram of studies investigating the effect of cytoreductive surgery (CRS) on the survival of endometrial cancer patients with peritoneal metastases. BMC: BioMed Central; MEDLINE: Medical Literature Analysis and Retrieval System Online.

**Figure 6 ijms-25-07495-f006:**
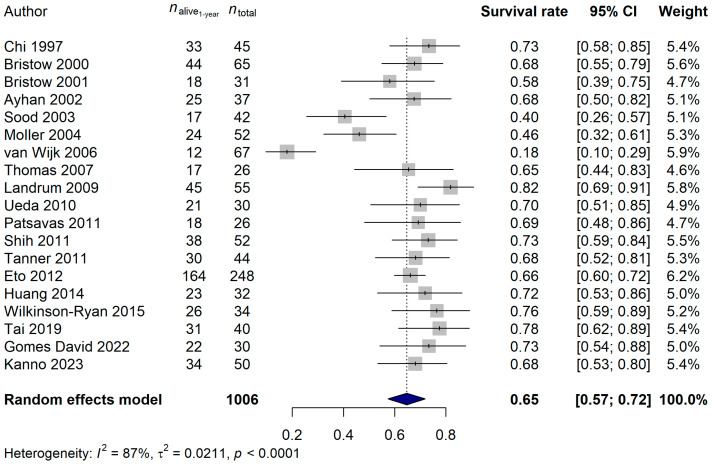
Effect of cytoreductive surgery (CRS) on the 1-year survival rate of endometrial cancer patients with peritoneal metastasis. Cited references: [7,34,35,36,37,38,39,40,41,42,43,44,45,46,47,48,50,52,53]. Whiskers, dotted line and blue diamond represents the 1-year survival rate and its 95% confidence interval (CI), the computed effect size, and the latter’s CI, respectively.

**Figure 7 ijms-25-07495-f007:**
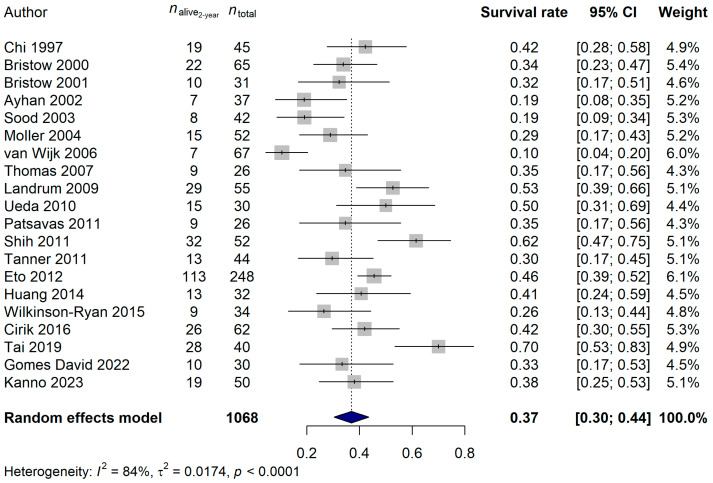
Effect of cytoreductive surgery (CRS) on the 2-year survival rate of endometrial cancer patients with peritoneal metastasis. Cited references: [7,34,35,36,37,38,39,40,41,42,43,44,45,46,47,48,50,51,52,53]. Whiskers, dotted line and blue diamond represents the 2-year survival rate and its 95% confidence interval (CI), the computed effect size, and the latter’s CI, respectively.

**Figure 8 ijms-25-07495-f008:**
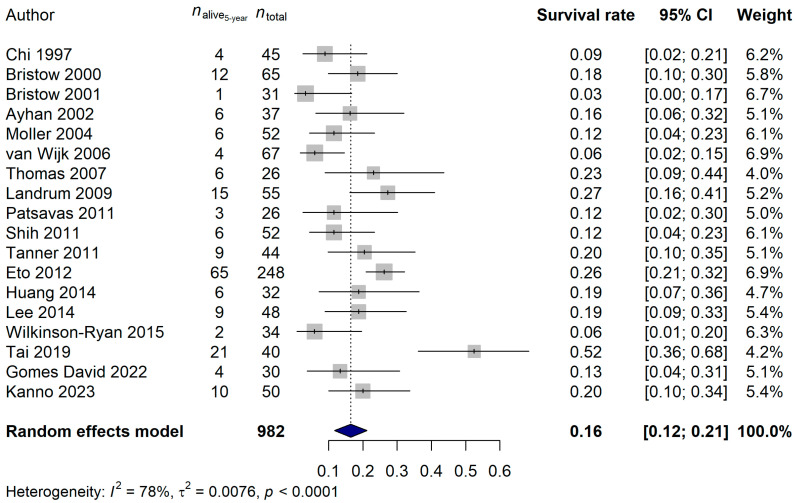
Effect of cytoreductive surgery (CRS) on the 5-year survival rate of endometrial cancer patients with peritoneal metastasis. Cited references: [7,34,35,36,37,39,40,41,42,44,45,46,47,48,49,50,52,53]. Whiskers, dotted line and blue diamond represents the 5-year survival rate and its 95% confidence interval (CI), the computed effect size, and the latter’s CI, respectively.

**Table 1 ijms-25-07495-t001:** Details of the selected studies investigating the effect of cytoreductive surgery (CRS) with hyperthermic intraperitoneal chemotherapy (HIPEC) on endometrial cancer with peritoneal metastasis.

Author (Year)	Type of Study	Cases (*n*)	Age (Mean)	PCI (Mean ± SD)	CRS + HIPEC Time (Hours; Mean)	HIPEC Time (min)	HIPEC Temperature (°C)	Hospital Stay Length (Days; Mean ± SD)
Helm et al. (2007) [25]	rObs	5	61	– ^1^	9.8	90	43	12.6 ± NA
Bakrin et al. (2010) [8]	rObs	5	55	9.60 ± 5.18	5.5	90	42	26.0 ± 27.4
Delotte et al. (2013) [26]	rObs	13	66.5	11.46 ± 5.93	5	60	43	13.4 ± 2.0
Abu-Zaid et al. (2014) [27]	rObs	6	55.5	19.33 ± 6.28	9.5	90	42	32.8 ± 9.5
Cornali et al. (2018) [28]	rObs	33	58	16.03 ± 10.67	6.25	60	43	18.6 ± 20.3
Navarro-Barrios et al. (2020) [32]	pObs	43	65	12.71 ± 9.2	7	60 / 90	42	11.0 ± 1.9
Brind’Amour et al. (2020) [29]	CS	3	59.7	15.00 ± 4.36	9.3	90	42	16.3 ± 7.4
Chambers et al. (2021) [30]	rObs	7	64.5	– ^1^	6.9	90	43	6.1 ± 3.7
Gomes David et al. (2022) [7]	rObs	30	60.9	9.86 ± 3.66	6.1	60 / 90	43	– ^1^
Rajha et al. (2022) [31]	CS	7	53	7.86 ± 3.98	4.55	60	42	16.4 ± 2.8

^1^ Not detailed in the original article. CS: case series; PCI: peritoneal carcinomatosis index; pObs: prospective observational study; rObs: retrospective observational study; SD: standard deviation.

**Table 2 ijms-25-07495-t002:** Details of the selected studies investigating the effect of cytoreductive surgery (CRS) in endometrial cancer with peritoneal metastasis.

Author (Year)	Type of Study	Cases (*n*)	Age (Mean)
Chi et al. (1997) [34]	rObs	45	66.6
Bristow et al. (2000) [35]	rObs	65	63.8
Bristow et al. (2001) [36]	rObs	31	63.8
Ayhan et al. (2002) [37]	rObs	37	61.2
Sood et al. (2003) [38]	rObs	42	65.9
Moller et al. (2004) [39]	rObs	52	67.1
van Wijk et al. (2006) [40]	rObs	67	62.3
Thomas et al. (2007) [41]	rObs	26	64.0
Landrum et al. (2009) [42]	rObs	55	62.5
Ueda et al. (2010) [43]	rObs	30	62.9
Patsavas et al. (2011) [44]	rObs	26	66.9
Shih et al. (2011) [45]	rObs	52	64.8
Tanner et al. (2011) [46]	rObs	44	68.4
Eto et al. (2012) [47]	rObs	248	59.0
Huang et al. (2014) [48]	rObs	32	61.8
Lee et al. (2014) [49]	rObs	48	70.0
Wilkinson-Ryan et al. (2015) [50]	rObs	34	69.1
Cirik et al. (2016) [51]	rObs	62	58.2
Tai et al. (2019) [52]	rObs	40	54.9
Gomes David et al. (2022) [7]	rObs	30	63.9
Kanno et al. (2023) [53]	rObs	50	60.2

rObs: retrospective observational study. Note: in comparison to Table 1, only a limited number of CRS-only studies provided data on peritoneal carcinomatosis index (PCI) and hospital stay length; therefore, these data are not provided here.

**Table 3 ijms-25-07495-t003:** Survival rates of the studies investigating cytoreductive surgery (CRS) with and without hyperthermic intraperitoneal chemotherapy (HIPEC) in endometrial cancer with peritoneal metastasis.

Parameter	All Studies		Studies Conducted after 2010	
	CRS only	CRS + HIPEC	CRS only	CRS + HIPEC
1-year survival rate (95% CI)	64.68% (57.42–71.94%)	82.28% (70.99–93.57%)	69.78% (66.07–73.48%)	81.86% (69.58–94.15%)
2-year survival rate (95% CI)	36.95% (30.39–43.52%)	56.07% (46.38–65.76%)	43.15% (35.27–51.03%)	54.05% (43.78–64.32%)
5-year survival rate (95% CI)	16.45% (11.74–21.17%)	21.88% (10.40–33.35%)	19.24% (12.12–26.36%)	21.88% (10.40–33.35%)

CI: confidence interval.

## Data Availability

The datasets used and/or analyzed during the current study are available from the corresponding author upon reasonable request.

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
