# Peer review of "Survival Difference of Endometrial Cancer Patients with Peritoneal Metastasis Receiving Cytoreductive Surgery (CRS) with and without Hyperthermic Intraperitoneal Chemotherapy (HIPEC): A Systematic Review and Meta-Analysis"

_ijms, 2024, doi:10.3390/ijms25137495_

Round 1

Reviewer 1 Report

Comments and Suggestions for Authors

 1.     In the CRS and CRS + HIPEC, did you include the laparoscopic & robotic approach methods?

2.     Did you compare the progression free survival (PFS) difference in these two groups?

3.     Is there any side effect or complication after adding of the HIPEC?

4.     What is the chemotherapy drug (and dosage) used in HIPEC?

5.     Is there any difference of residual tumor status after operation between these two groups?

6.     Since the 5-year survival in not significantly different in these two groups, is there any significant benefit for patient of the improved only 1- and 2- year survival?

Author Response

We thank Reviewer 1 for the positive feedback on our article. Our answers for the questions and critical comments are below:

  1. In the CRS and CRS + HIPEC, did you include the laparoscopic & robotic approach methods?

Reply: Thank you for Your kind question. All reviewed literature (both CRS and CRS+HIPEC studies) only included cases, where the classic laparoscopic approach was applied to remove the infiltrated endometrial tissue. No robotic approach methods were reported.

  1. Did you compare the progression free survival (PFS) difference in these two groups?

Reply: Unfortunately, the data on PFS could not be analyzed due to the fact that PFS was not only not provided in multiple studies, but its definition significantly varied among them. Moreover, in most of the studies, PFS was not provided for the whole cohort, but was used only for a specific comparison, such as CC0 vs CC1+, etc. Therefore, we were unable to statistically analyze the data and therefore include it in our meta-analysis.

  1. Is there any side effect or complication after adding of the HIPEC?

Reply: Thank you for the hint. The following was included in the Discussion.

“According to current literature, the addition of HIPEC is not associated with signifi-cantly higher rates of complications [doi: 10.1007/s10585-019-09970-5 and 10.1186/s12893-021-01449-z.]. Most of its side-effects originate from the used chemotherapeutic agent. E.g., the most common side effect of platinum-containing intraperitoneal chemotherapy is the renal toxicity, which can be avoided by the use of sodium thiosulfate. Furthermore, hematologic side-effects such as inhibition of hematopoiesis and neutropenia due to bone marrow suppression are common after HIPEC [doi: 10.7717/peerj.15277]. Nevertheless, most of the complications are more likely to be attributed to the extensive cytoreduction and not to the additive HIPEC. In a recent multi-institutional comparative study from PSOGI and RENAPE groups [doi: 10.1186/s12893-021-01449-z], where 30 patients were compared (CRS with HIPEC vs. CRS alone) the Grade III and IV complications according the Clavien–Dindo classification rates did not significantly differ between the CRS plus HIPEC and CRS only group (20.7% vs. 20.7%, p = 0.739). In this study, one patient (3.3%) in each group experienced abdominal hemorrhage and required blood transfusion. The most frequent complication was gastrointestinal complication (overall 11.8% of patients), which wasn’t further specified. Another study conducted by Cornali et al. [doi: 10.1245/s10434-017-6307-3] reported a Clavien-Dindo I-II, III, IV and V complication rate of 33.3%, 15.1%, 3% and of 3% of the included 33 patients, respectively.”

  1. What is the chemotherapy drug (and dosage) used in HIPEC?

Reply: Thank you for this important question. Our review includes the following statement regarding the used chemotherapeutic agents and their dosage: “Due to the fact that HIPEC is still evolving as a treatment modality, no consensus has been reached yet on subjects such as the optimal temperature to be used (41 °C vs. 42 °C vs. 43 °C), the length of HIPEC (60 vs. 90 min), and the chemotherapeutic agent used during the procedure. In all of the reviewed studies, the HIPEC temperature varied between 42 °C and 43 °C. HIPEC length was either 60 or 90 minutes long but there were studies, in which the authors used both 60– and 90-minute-long procedures. The chemotherapeutic agents used during the procedure also had a wide variability (cisplatin, doxorubicin, mitomycin C, and oxaliplatin).”

Here we provide additional data that was extracted from the studies regarding the used chemotherapy drugs and their dosage (if provided). This data was originally omitted due to the better visibility of Table 1. In the revised manuscript, the below table was introduced as Table S1.

Author (Year)

Cases

(n)

CRS + HIPEC time (hours; mean)

HIPEC time (min)

HIPEC

temperature (°C)

HIPEC medication

Helm et al. (2007) [25]

5

9.8

90

43

cisplatin

Bakrin et al. (2010) [8]

5

5.5

90

42

cisplatin (0,7 mg/kg)

mitomycin C (0,5mg/kg)

Delotte et al. (2013) [26]

13

5

60

43

cisplatin (50 mg/m2)

doxorubicin (15 mg/m2)

Abu-Zaid et al. (2014) [27]

6

9.5

90

42

cisplatin (50 mg/m2)

doxorubicin (15 mg/m2)

Cornali et al. (2018) [28]

33

6.25

60

43

cisplatin (75 mg/m2)

Navarro-Barrios et al. (2020) [32]

43

7

60 / 90

42

cisplatin, paclitaxel

Brind’Amour et al. (2020) [29]

3

9.3

90

42

carboplatin

Chambers et al. (2021) [30]

7

6.9

90

43

cisplatin (100 mg/m2)

paclitaxel (135 – 175 mg/m2)

Gomes David et al. (2022) [7]

30

6.1

60 / 90

43

cisplatin, doxorubicin, mitomycin C

Rajha et al. (2022) [31]

7

4.55

60

42

cisplatin, doxorubicin

  1. Is there any difference of residual tumor status after operation between these two groups?

Reply: During our analysis of the literature, we collected data regarding the status of residual tumor after operation in the format of Completeness of Cytoreduction score (CC-score). Unfortunately, all studies interpreted CC-scores in different ways. While some studies provided details on the exact CC scores achieved after surgery, most studies only reported whether the debulking was “optimal” or “suboptimal”, without providing specific, exact or homogenous details. In turn we were unable to provide statistics regarding the difference between CRS and CRS+HIPEC in terms of the residual tumor status post-surgery. This issue was detailed in the Discussion.

  1. Since the 5-year survival is not significantly different in these two groups, is there any significant benefit for patient of the improved only 1- and 2- year survival?

Reply: Thank you very much for your important comment. To date there is no randomized controlled trial in these patients and clinical setting. Furthermore, the recently introduced immunotherapy wasn’t part of the multimodal therapy so far, which could possibly serve as a “boost” with synergistic effect in the future. Nevertheless, the 2-year survival rate varies between 10 and 70% in the studies, in a disease situation with a very poor prognosis. The addition of HIPEC still offers the best chances of survival according to the current meta-analysis. Besides unfortunately, we still lose indeed many patients over the years.

Reviewer 2 Report

Comments and Suggestions for Authors

This paper investigates the possible role in terms of additional survival benefits of cytoreductive surgery (CRS) with hyperthermic intraperitoneal chemotherapy (HIPEC) over cytoreduction alone in the treatment of patients with advanced-stage endometrial cancer with peritoneal metastases.

The paper is well written and the English language is appropriate and understandable.

The clinical topics are fascinating. The management of endometrial cancer is rapidly evolving because of the impact of new molecular assessment and classification on endometrial cancers.

The rationale behind using HIPEC after cytoreductive surgery is the association between the pharmacological activity of chemotherapy delivered to the peritoneal cavity and the enhanced cytotoxic effect of hyperthermia.

So far, data on the efficacy of HIPEC are still controversial and limited by a limited number of trials (all of them are non-randomized), the small sample size of most of the studies, and the inclusion of different treatment settings and chemotherapy regimens.

Due to some criticisms raised about the results of a few published studies, currently, many authors do not agree to include HIPEC in daily clinical practice until more data on its efficacy and safety from randomized control trials are provided.

Even though this paper is the first to analyze CRS alone and CRS plus HIPEC study results of endometrial cancer patients with peritoneal metastases in a single meta-analysis, the conclusions are misleading. The Authors report that the addition of HIPEC to CRS can significantly increase the positive clinical outcome, especially in the first  2 years. However, analyzing papers published in the last decade to reduce the high heterogeneity of the included studies, no statistically significant difference is observed.

Could the Authors provide data regarding histotype, adjuvant treatment (chemotherapy alone versus combined chemotherapy plus radiotherapy) residual disease after CRS, and the presence of pelvic and aortic bulky nodes? 

Could the Authors report at least a few data on complications?

Table 2 should show data on length of hospitalization.

Author Response

We thank Reviewer 2 for the positive feedback on our article. Our answers for the critical comments are below:

This paper investigates the possible role in terms of additional survival benefits of cytoreductive surgery (CRS) with hyperthermic intraperitoneal chemotherapy (HIPEC) over cytoreduction alone in the treatment of patients with advanced-stage endometrial cancer with peritoneal metastases. The paper is well written and the English language is appropriate and understandable. The clinical topics are fascinating. The management of endometrial cancer is rapidly evolving because of the impact of new molecular assessment and classification on endometrial cancers. The rationale behind using HIPEC after cytoreductive surgery is the association between the pharmacological activity of chemotherapy delivered to the peritoneal cavity and the enhanced cytotoxic effect of hyperthermia. So far, data on the efficacy of HIPEC are still controversial and limited by a limited number of trials (all of them are non-randomized), the small sample size of most of the studies, and the inclusion of different treatment settings and chemotherapy regimens. Due to some criticisms raised about the results of a few published studies, currently, many authors do not agree to include HIPEC in daily clinical practice until more data on its efficacy and safety from randomized control trials are provided.

  1. Even though this paper is the first to analyze CRS alone and CRS plus HIPEC study results of endometrial cancer patients with peritoneal metastases in a single meta-analysis, the conclusions are misleading. The Authors report that the addition of HIPEC to CRS can significantly increase the positive clinical outcome, especially in the first 2 years. However, analyzing papers published in the last decade to reduce the high heterogeneity of the included studies, no statistically significant difference is observed.

Reply: We thank the Reviewer for their kind criticism. In light of that, the abstract, discussion and conclusion of the manuscript had been revised with a more moderate claim.

  1. Could the Authors provide data regarding histotype, adjuvant treatment (chemotherapy alone versus combined chemotherapy plus radiotherapy) residual disease after CRS, and the presence of pelvic and aortic bulky nodes?

Reply: Thank you for your comment. Due to the extremely high heterogeneity of both the used adjuvant therapy and the specific histological types of the malignancies, this data was not included in our meta-analysis. Due to the fact, that most studies reported these data differently, none of these data could be converted to a single format, with which any meta-comparisons would be feasible. Some additional, general data is provided below.

From the analyzed studies it can be reported that the most common histology was endometrioid adenocarcinoma (57%) followed by serous high-grade carcinoma (36%). The minority included carcinosarcomas (3%), adenosquamous (0,4%) and clear cell carcinomas (3%). In terms of the adjuvant treatment, we observed that most commonly post operative chemotherapy or chemoradiotherapy was implemented. Regarding the residual tumor status, while some studies provided details on the exact CC scores achieved after surgery, most studies only reported whether the debulking was “optimal” or “suboptimal”, without providing specific, exact or homogenous details. In turn we were unable to provide statistics regarding the difference between CRS and CRS+HIPEC in terms of the residual tumor status post-surgery. The presence of bulky lymph nodes in both the pelvic and aortic region is mentioned in very few papers from the cohort, meaning once again we are unable to provide information regarding the question. 

  1. Could the Authors report at least a few data on complications?

Reply: Thank you for your comment.  According to current literature, the addition of HIPEC is not associated with significantly higher rates of complications. (doi: 10.1007/s10585-019-09970-5, doi: 10.1186/s12893-021-01449-z.) The complications are more likely to be attributed to the extensive cytoreduction and not to the additive HIPEC. In a recent multi-institutional comparative study from PSOGI and RENAPE groups (doi: 10.1186/s12893-021-01449-z), where 30 patients were compared (CRS with HIPEC vs. CRS alone) the Grade III and IV complications according the Clavien–Dindo classification rates did not significantly differ between the CRS plus HIPEC and CRS only group (20.7% vs 20.7%, p = 0.739). In this study, one patient (3.3%) in each group experienced abdominal hemorrhage and required blood transfusion. The most frequent complication was gastrointestinal complication (overall 11.8% of patients), which wasn’t further specified. Another study conducted by Cornali et al. reported a Clavien-Dindo I-II, III, IV and V complication rate of 33.3%, 15.1%, 3% and of 3% at 33 patients, respectively, whereas the individual groups were not specified.

  1. Table 2 should show data on length of hospitalization.

Reply: Thank you for your comment. Unfortunately, due to the fact that the vast majority of the reviewed literature did not provide data on the length of the hospitalization in the case of CRS-only studies, we were unable to provide further details regarding this data. A note about this issue had been added to the footer of Table 2.

Reviewer 3 Report

Comments and Suggestions for Authors

The article titled "Survival Difference of Endometrial Cancer Patients with Peritoneal Metastasis Receiving Cytoreductive Surgery (CRS) with and without Hyperthermic Intraperitoneal Chemotherapy (HIPEC): a Systematic Review and Meta-analysis" by Panczel, et al. includes a systematic review and meta-analysis from the current literature on the cytoreductive surgery and its effects/complications in advanced Endometrial Cancer.

The study is well-organized and written. The study results are presented clearly and the statistical analysis is appropriately performed. It consists of a comprehensive study of the literature on the effect of CRS alone on the survival of ECPM, the effect of CRS+HIPEC on the survival of ECPM, and the comparison of these two. The systematic review is very well written, including also the cellular and molecular effects of HIPEC.

The strength of the study is that there is no other meta-analysis to compare the survival rates at 1 year, 2 years, and 5 years between CRS only and CRS+HIPEC, the results being in favor of the latter, except at 5 years (there was no statistically significant difference).

In conclusion, taking into consideration all the above mentioned I consider the current article is an important contribution to the literature.  

Author Response

We thank Reviewer 3 for your work and time.

The article titled "Survival Difference of Endometrial Cancer Patients with Peritoneal Metastasis Receiving Cytoreductive Surgery (CRS) with and without Hyperthermic Intraperitoneal Chemotherapy (HIPEC): a Systematic Review and Meta-analysis" by Panczel, et al. includes a systematic review and meta-analysis from the current literature on the cytoreductive surgery and its effects/complications in advanced Endometrial Cancer.

The study is well-organized and written. The study results are presented clearly and the statistical analysis is appropriately performed. It consists of a comprehensive study of the literature on the effect of CRS alone on the survival of ECPM, the effect of CRS+HIPEC on the survival of ECPM, and the comparison of these two. The systematic review is very well written, including also the cellular and molecular effects of HIPEC.

The strength of the study is that there is no other meta-analysis to compare the survival rates at 1 year, 2 years, and 5 years between CRS only and CRS+HIPEC, the results being in favor of the latter, except at 5 years (there was no statistically significant difference).

In conclusion, taking into consideration all the above mentioned I consider the current article is an important contribution to the literature.

Reply: Thank you very much and the consideration to publish our investigation in in International Journal of Molecular Sciences.